# Hematopoietic chimerism and donor-specific skin allograft tolerance after non-genotoxic CD117 antibody-drug-conjugate conditioning in MHC-mismatched allotransplantation

Zhanzhuo Li[1], Agnieszka Czechowicz[2,3,4,5,6,7,8], Amelia Scheck[2,3,4,5,7,8], Derrick J. Rossi 🄳 [2,3,4,5] & Philip M. Murphy[1]

Hematopoietic chimerism after allogeneic bone marrow transplantation may establish a state of donor antigen-specific tolerance. However, current allotransplantation protocols involve genotoxic conditioning which has harmful side-effects and predisposes to infection and cancer. Here we describe a non-genotoxic conditioning protocol for fully MHC-mismatched bone marrow allotransplantation in mice involving transient immunosuppression and selective depletion of recipient hematopoietic stem cells with a CD117-antibody-drug-conjugate (ADC). This protocol resulted in multilineage, high level (up to 50%), durable, donor-derived hematopoietic chimerism after transplantation of 20 million total bone marrow cells, compared with ≤ 2.1% hematopoietic chimerism from 50 million total bone marrow cells without conditioning. Moreover, long-term survival of bone marrow donor-type but not third party skin allografts is achieved in CD117-ADC-conditioned chimeric mice without chronic immunosuppression. The only observed adverse event is transient elevation of liver enzymes in the first week after conditioning. These results provide proof-of-principle for CD117-ADC as a non—genotoxic, highly-targeted conditioning agent in allotransplantation and tolerance protocols.

---

[1] Laboratory of Molecular Immunology, National Institute of Allergy and Infectious Diseases (NIAID), National Institutes of Health, Bethesda 20892 MD, USA. [2] Program in Cellular and Molecular Medicine, Boston Children's Hospital, Boston 02115 MA, USA. [3] Department of Stem Cell and Regenerative Biology, Harvard University, Cambridge 02138 MA, USA. [4] Division of Hematology/Oncology, Department of Pediatrics, Harvard Medical School, Boston 02115 MA, USA. [5] Harvard Stem Cell Institute, Cambridge 02138 MA, USA. [6] Department of Pediatric Oncology, Dana Farber Cancer Institute, Boston 02115 MA, USA. [7] Department of Pediatrics, Division of Stem Cell Transplantation and Regenerative Medicine, Stanford University School of Medicine, Stanford 94304 CA, USA. [8] Institute for Stem Cell Biology and Regenerative Medicine, Stanford University School of Medicine, Stanford 94305 CA, USA. These authors contributed equally: Zhanzhuo Li, Agnieszka Czechowicz. These authors jointly supervised this work: Derrick J. Rossi, Philip. M. Murphy. Correspondence and requests for materials should be addressed to A.C. (email: aneeshka@stanford.edu) or to D.J.R. (email: derrick.rossi@childrens.harvard.edu) or to P.M.M. (email: pmm@nih.gov)

Monoclonal antibody (mAb)-based approaches for depleting recipient hematopoietic stem cells (HSCs) have shown promise as non-genotoxic conditioning agents in bone marrow (BM)/HSC transplantation (BMT/HSCT)[1–6]. mAb targeting of CD117 (c-Kit)[7], a receptor tyrosine kinase that is highly expressed on HSCs and that binds the cytokine stem cell factor (SCF), was first shown to enhance HSC engraftment after syngeneic HSCT in immunodeficient mice; however, this stand-alone approach was unsuccessful in adult wild-type, immunocompetent mice[2]. Subsequent improvements have included (1) combining an antagonistic anti-CD117 mAb with CD47 blockade, which promoted engraftment after syngeneic BMT and allogeneic BMT across a minor MHC mismatch[5], and (2) saporin−conjugated anti-CD45.2 immunotoxin conditioning, which achieved robust syngeneic chimerism in immunocompetent animals but was never shown to be effective in allogeneic settings[6].

Given the limitations with prior methods, we have developed a novel saporin−based CD117 antibody-drug-conjugate (CD117-ADC) that as a single-agent potently and selectively depletes recipient HSCs without immune or hematopoietic ablation and supports robust (~99%) and long-term (>1 year) hematopoietic chimerism after syngeneic BMT and HSCT in adult, immunocompetent mice without limiting morbidity or mortality[8]. This approach has obvious advantages for syngeneic applications in the clinic where preservation of immunity is desired, such as autologous gene therapy and gene editing.

Here we extend this approach to allotransplantation and show that conditioning with CD117-ADC and transient immunosuppression safely promotes robust hematopoietic chimerism with durable donor-specific skin allograft tolerance in the setting of fully MHC-mismatched allotransplantation. Using this approach, we observe no graft versus host disease or other limiting toxicity. Hematopoietic chimerism is achieved with relatively low numbers of transplanted bone marrow cells and reaches levels compatible with those needed for reversing the phenotype of many grievous genetic diseases of the blood such as sickle cell disease and chronic granulomatous disease[9,10]. Moreover, as expected, chimeric animals reach a state of donor-specific tolerance as defined by persistent survival of donor-type skin allografts without need for further immunosuppression. The strong cell-sparing effect, lack of genotoxicity and robust donor-specific tolerance associated with the protocol establish a pre-clinical proof-of-principle for the use of HSC-depleting antibodies such CD117-ADC as safe and effective conditioning agents for allotransplantation.

## Results

To test the safety and efficacy of CD117-ADC in the allogeneic setting, we performed sequential, fully MHC-mismatched BMT and skin transplantation in mice with BALB/c donors and C57Bl/6 recipients. Recipients were conditioned once with CD117-ADC treatment 6 days before BMT, and then given transient immunosuppression as per a previous protocol for MHC-mismatched transplantation to prevent acute graft rejection (one dose each of depleting anti-CD8 mAb, and non-depleting anti-CD4 and anti-CD154 mAbs on days 0, +2, and +4, plus rapamycin on days +6 and +30)[11] (Fig. 1a). Subsequently, transplantation of tail skin from BALB/c mice (BM donor) and CBA/Ca mice (a genetically and immunologically distinct third-party donor) was performed contemporaneously on BMT-recipient C57Bl/6 mice twice, ~5 (primary allografts) and 8 (secondary allografts) months after BMT (Fig. 1a). Without pre-transplant conditioning, $\geq 5 \times 10^7$ donor BM cells are required in this model to establish de minimis hematopoietic chimerism (1–2%), which reliably establishes donor-specific skin allograft tolerance[12,13].

**CD117-ADC conditioning promotes hematopoietic chimerism.** Remarkably, a single dose of CD117-ADC conditioning before $2 \times 10^7$ BALB/c BM cell transplantation in combination with transient immunosuppression resulted in high levels of hematopoietic chimerism in 14 of 15 C57Bl/6 recipients (Fig. 1b) across this complete MHC-mismatch. Robust chimerism lasted the entire observation period (411–624 days) in 13 of the 14 chimeric mice (Fig. 1c and Supplementary Figure 1). Only low-level chimerism (0.44–2.14%) occurred after transplanting $5 \times 10^7$ BALB/c BM cells without conditioning (Fig. 1b and Supplementary Figure 1), confirming previous reports[12,13]. Chimerism was not observed in mice receiving $2 \times 10^7$ BALB/c BM cells when unconditioned or conditioned with unconjugated anti-CD117 mAb or saporin-conjugated isotype control antibody (ISO-ADC) (Fig. 1b and Supplementary Figure 1). There appeared to be some waning of total donor chimerism in the peripheral blood, but only in a minority of animals and only very late after transplantation. It is important to note that the transplanted mice were followed for almost 2 years post-transplantation (Fig. 1c), near their life expectancy, and that all but one of those who developed chimerism in the blood were still chimeric at the time of experiment termination. Additionally, even those with the lowest levels of mixed chimerism in the blood at the endpoint still had high levels of chimerism in secondary lymphoid organs (Fig. 1d). In particular, on termination 411–624 days post BMT, high levels of donor chimerism were also observed in CD117-ADC-conditioned but not ISO-ADC-conditioned recipients in BM, spleen, axillary and mesenteric lymph nodes, Peyer's patch, liver and lung (Fig. 1d). High levels of donor chimerism were observed in CD117-ADC-conditioned recipients for all mature leukocyte subsets tested (CD3+ T cells, CD11b+ monocytes, CD19+ B cells, CD49b+ NK cells, and Gr1+ neutrophils) as well as Ter119+ erythroid progenitors in both BM and spleen, with B cells achieving the highest level of chimerism in both compartments (Fig. 1e). CD117-ADC conditioning induced sustained high-level donor chimerism in the blood for erythrocytes and all mature leukocyte subsets tested (Supplementary Figure 2).

At killing 411–624 days after complete MHC-mismatched BMT, levels of donor stem and progenitor cell chimerism for CD117-ADC-conditioned mice in the BM were $3.4 \pm 0.76\%$ (mean ± SEM, $n = 7$) for long-term HSCs, $6.7 \pm 1.99\%$ for short-term HSCs and $2.9 \pm 0.87\%$ for multipotent progenitors[14], whereas chimerism was not observed for these cell types ($<0.41 \pm 0.2\%$, $n = 3$) in BM from ISO-ADC-conditioned animals (Fig. 1f). Importantly, we observed no significant weight loss or evidence of graft versus host disease after CD117-ADC-conditioned allogeneic BMT. In addition, at the time of killing the total leukocyte count in peripheral blood of mice receiving CD117-ADC was $11.07 \pm 1.54$ K per μL, similar to ISO-ADC-conditioned mice ($9.18 \pm 0.4$ K per μL). As in the syngeneic bone marrow transplantation model[8], we observed that CD117-ADC conditioning in the allotransplantation setting did not cause any gross toxicity and mice appeared healthy with preserved coat color post treatment, although similar transient elevations of liver function tests were observed (Supplementary Figure 3). Since CD117 is not just expressed on hematopoietic stem and progenitor cells, but also on mast cells, melanocytes, and on other rare cells including some intestinal cells and germ cells, once a clinical immune suppression regimen is optimized, further experiments will be needed to examine whether CD117-ADC conditioning in the allogeneic setting affects the function of these and other CD117 expressing cells. However, when used as a sole treatment no gross effects have been observed to date[8].

**CD117-ADC conditioning promotes skin allograft tolerance.** Given immunologic MHC-mismatch, both primary and

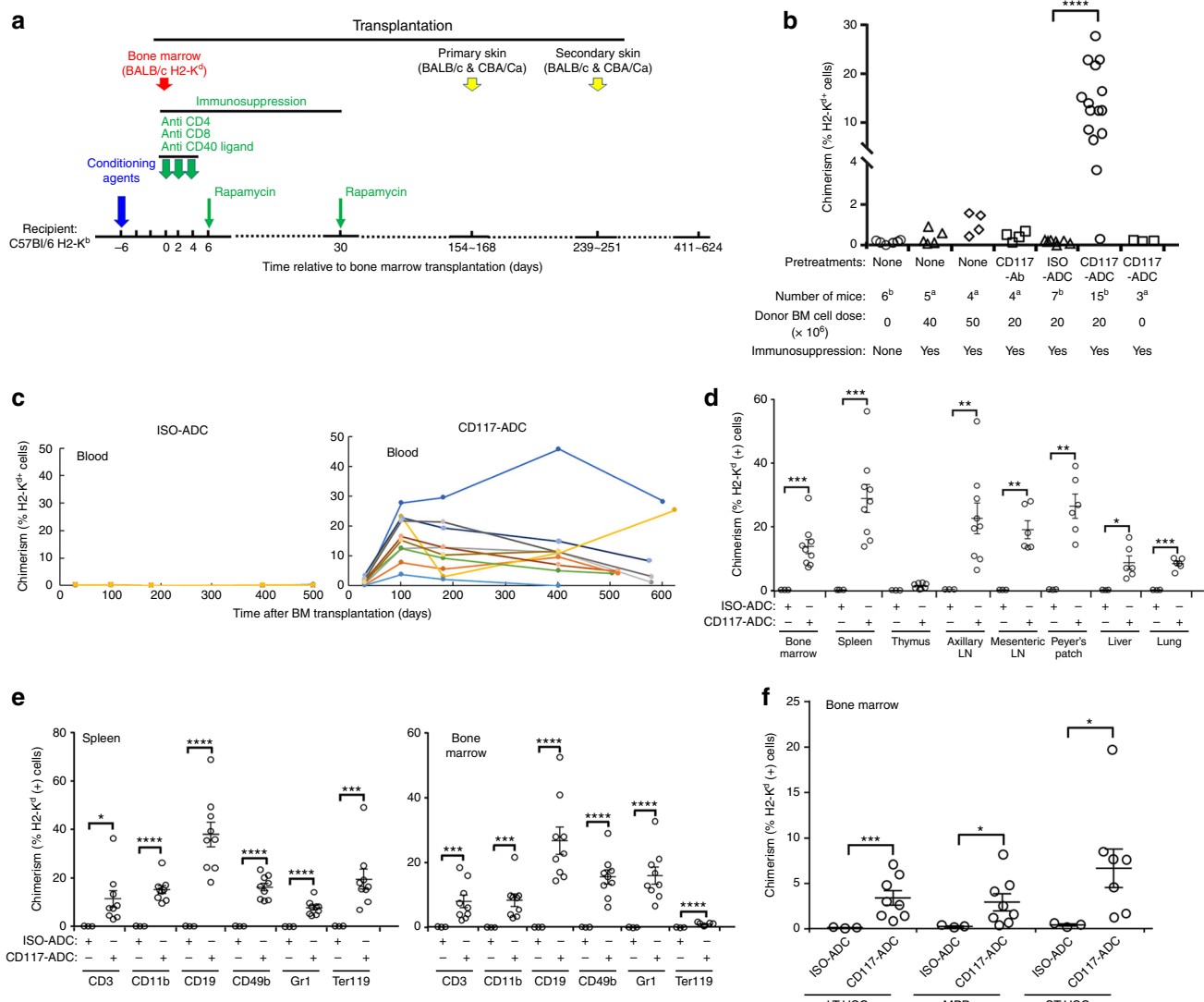

**Fig. 1** Robust hematopoietic chimerism after CD117-ADC conditioning and transient immunosuppression in fully MHC-mismatched BM allotransplantation. **a** Experimental protocol. **b**–**f** Abbreviations: *ADC*, antibody-drug-conjugate; *CD117*, anti-CD117 mAb; *ISO*, isotype control mAb. **b** Donor peripheral blood chimerism showing CD117-ADC efficacy. Data are from day 100 post BMT and are summarized as the mean ± SEM from one ([a]) or two ([b]) independent experiments. CD117-Ab, 2 animals each for unconjugated CD117 mAbs 2B8 and ACK2. The underlying FACS gating strategy is given in Supplementary Figure 4. **c** Engraftment time course. Each line represents an individual mouse from a single experiment. The underlying FACS gating strategy is given in Supplementary Figure 4. **d** Immune organ chimerism. The underlying FACS gating strategy is given in Supplementary Figure 5. **e** Chimerism of mature leukocytes in spleen and BM. The underlying FACS gating strategy is given in Supplementary Figure 6. **f** BM progenitor chimerism. LT-HSC, long-term-HSCs (Lin⁻Sca1+c-Kit+CD34^{lo}Flt3^{lo}); ST-HSCs, short-term-HSCs (Lin⁻Sca1+c-Kit+CD34^{hi}Flt3^{lo}); MPP, multipotent progenitors (Lin⁻Sca1+c-Kit +CD34^{hi}Flt3^{hi}). The underlying FACS gating strategy is given in Supplementary Figure 7. **b, d**–**f** Data are the mean ± SEM donor chimerism at termination > 500 days after BMT. Statistics calculated using unpaired parametric *t*-tests with Welch's correction (two-tailed); all data points significant as indicated (*$P < 0.05$; **$P < 0.01$; ***$P < 0.001$; ****$P < 0.0001$). **c**–**f** Results are from one experiment representative of two independent experiments with 4–9 animals in each group

secondary third-party CBA/Ca skin allografts were rapidly rejected (in < 20 days) in all groups tested (Fig. 2a, b). Donor-type BALB/c primary skin allografts were also rapidly rejected in C57BL/6 mice transplanted with $2 \times 10^7$ BALB/c BM cells after conditioning with ISO-ADC. In contrast, both primary and secondary skin allografts survived in 13 of 15 mice transplanted with CD117-ADC conditioning for up to 412 days post-primary skin allotransplantation and for at least 230 days after secondary skin transplantation (Fig. 2a, b), when the animals were arbitrarily terminated to evaluate immune organ chimerism. Importantly, the two exceptions which both sustained early primary skin allograft rejection before receiving secondary allografts

had failed to develop or maintain significant hematopoietic chimerism after BMT for unclear reasons potentially due to poor injections (Fig. 1b, c).

Thus, it is apparent that CD117-ADC pre-treatment fosters establishment of a state of hematopoietic chimerism post BMT even across complete MHC-mismatch which promotes a state of clinical tolerance as defined by (1) long-term survival of a well-functioning allograft without immunosuppression in an immunocompetent individual and (2) in vivo unresponsiveness to BMT donor-type but not third-party organs, tissues and cells. Nevertheless, histological analysis conducted at 160 to 456 days after skin allotransplantation and 411–624 days after bone

**a**

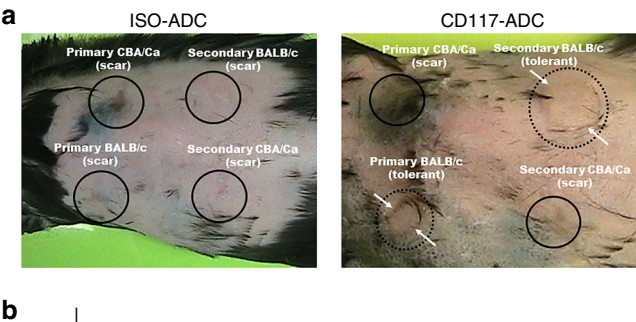

**b**

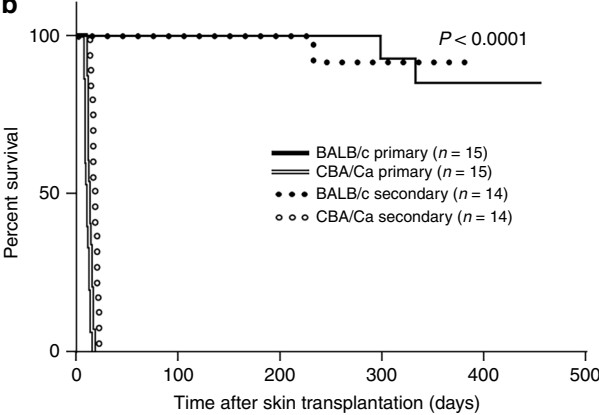

**Fig. 2** Robust donor-specific tolerance after CD117-ADC conditioning and fully MHC-mismatched sequential bone marrow and skin allotransplantation. C57Bl/6 animals were conditioned as indicated at the top of each panel, then transplanted with $2 \times 10^7$ total BM cells, followed by skin allotransplantation, according to the protocol shown in Fig. 1a. **a** Representative example of skin allograft outcome 336 (primary) and 265 (secondary) days post skin transplantation. BM donor-type (BALB/c) and third-party (CBA/Ca) skin allografts were transplanted at 168 (primary) and 239 (secondary) days post BMT preceded by conditioning with isotype control antibody (left) or CD117-ADC (right). Circles locate the site of skin transplantation: solid lines, rejected transplants; dotted lines, viable transplants. Arrows demarcate the margin of the surviving skin allograft. **b** Skin allograft survival curves summarized from two independent experiments comparing conditioning regimens. The survival was analyzed by the following tests: Log-rank (Mantel-Cox) test ($\chi 2$ 65.88), Log-rank test for trend ($\chi 2$ 25.75), and Gehan-Breslow-Wilcoxon test ($\chi 2$ 56.36)

marrow transplantation revealed mononuclear cell infiltration in the deep dermis of the surviving primary and secondary donor-type BALB/c tail skin allografts, in significant contrast to adjacent C57Bl/6 recipient skin and naïve non-transplanted BALB/c tail skin (Fig. 3a). Comparison to transplanted non-donor type CBA/Ca tail skin allotransplants was not possible since the transplants had been acutely rejected and the transplant site had healed with recipient skin (Fig. 2a, b). Compared to control skin, the epidermis of the donor-type BALB/c tail skin allografts was slightly thickened but otherwise intact and non-infiltrated. Furthermore, the level of collagen content in the dermis, as a measure of fibrosis, assessed by Masson's trichrome stain, was similar in the donor-type BALB/c tail skin allografts and the non-transplanted control BALB/c tail skin (Fig. 3a). Subsequent investigation of the cell types infiltrating the allograft dermis by immunohistochemistry revealed CD45 (+) hematopoietic cells, including rare CD3 (+) T cells, but greater numbers of B220 (+) B cells and F4/80 (+) macrophages (Fig. 3b). The pathophysiologic significance of these cells is unknown given that they do not appear to be mediating rejection as evidenced by the histologically intact allografts and non-infiltrated epidermis in all animals

>160 days after skin allotransplantation and could contain regulatory and/or pro-inflammatory subsets of each cell type. However, their presence does raise the question whether the third criterion for tolerance, i.e. absence of histologic evidence of rejection, has been fully achieved with this CD117-ADC conditioning, transient immunosuppression and bone marrow transplantation protocol and provides an opportunity to fully explore the molecular and cellular mechanisms of tolerance in an irradiation-free system. Although the possibility must also be considered that these infiltrating leukocytes might represent donor cells engaged in incipient graft versus host disease, this is unlikely since these cells were only infiltrating the donor skin allografts, not the surrounding recipient skin (Fig. 3c).

## Discussion

Obviating up-front genotoxic conditioning and enabling MHC-mismatched allotransplantation with induction of allo-tolerance are major unmet medical needs in the transplantation field that for the first-time appear to be satisfied with a targeted, HSC-depleting agent (CD117-ADC). The favorable safety profile of CD117-ADC may relate to the selectivity of the agent for CD117, which is expressed at high levels on few cell types besides HSCs (Expression Atlas, The European Bioinformatics Institute (EMBL-EBI), CD117, accessed 6 December 2018. https://www.ebi.ac.uk/gxa/). CD117-ADC conditioning has the virtue of being simple and scalable, and is potentially amenable to enhanced efficacy by further optimization of the transient immunosuppression protocol used. It also has the virtue of potential broad applicability in both hematopoietic allotransplantation and allogeneic solid organ transplantation. In this regard, the level of CD117-ADC-conditioned hematopoietic chimerism observed in our study is already sufficient to cure many inherited diseases of the blood[15–17]. In particular, clinical manifestations of severe hemoglobinopathies and chronic granulomatous disease may be attenuated or abolished by relatively low levels of erythrocyte and neutrophil chimerism, respectively, that were attained in our proof of concept allo-transplantation study with CD117-ADC conditioning. If successfully applied to solid organ allotransplantation, CD117-ADC conditioning might additionally allow long-term survival of allografts without chronic immunosuppression post transplantation, potentially reducing acute and chronic morbidity and mortality[18–20], and potentially increasing the supply of organ donors if tolerance is achieved and MHC-matching is not required. Additional work will be needed to optimize the approach by systematically studying the immunosuppression regimen, dosing schedule, the number of donor cells transplanted, the age at transplantation, and different CD117 antibodies and drug-conjugates, among other parameters. Clinical translation is likely given that several therapeutic anti-human CD117 antibodies and ADCs are already under advanced clinical development[21–23]. In conclusion, this CD117-ADC system illustrates that antibody-based conditioning can be used to enable complete MHC-mismatched transplantation which could have broad implications for transplantation across many settings.

## Methods

**Antibody-drug-conjugate preparation and administration**. Biotinylated monoclonal antibodies directed against mouse CD117/c-kit (clone 2B8, Biolegend, San Diego, CA; Lot # 170777–31) were coupled to streptavidin-linked saporin toxin (Advanced Targeting systems, San Diego, CA; Lot # 94-31) as detailed in Czechowicz et al.[8] leading to generation of CD117-antibody-drug conjugate (CD117-ADC). Each mouse was injected with 1.5 mg/kg of ADC (~12 μg of streptavidin-saporin) in a total volume of 300 μl of PBS as was previously optimized in Palchaudhuri et al.[6] and Czechowicz et al.[8]. A biotinylated rat IgG2b kappa isotype control antibody (RTK4530, Biolegend, San Diego, CA) conjugated to saporin and mouse CD117/c-kit-specific antibodies (clones 2B8 & ACK2) unconjugated to saporin served as controls and were dosed at 500 μg per animal (~25 mg per kg) as

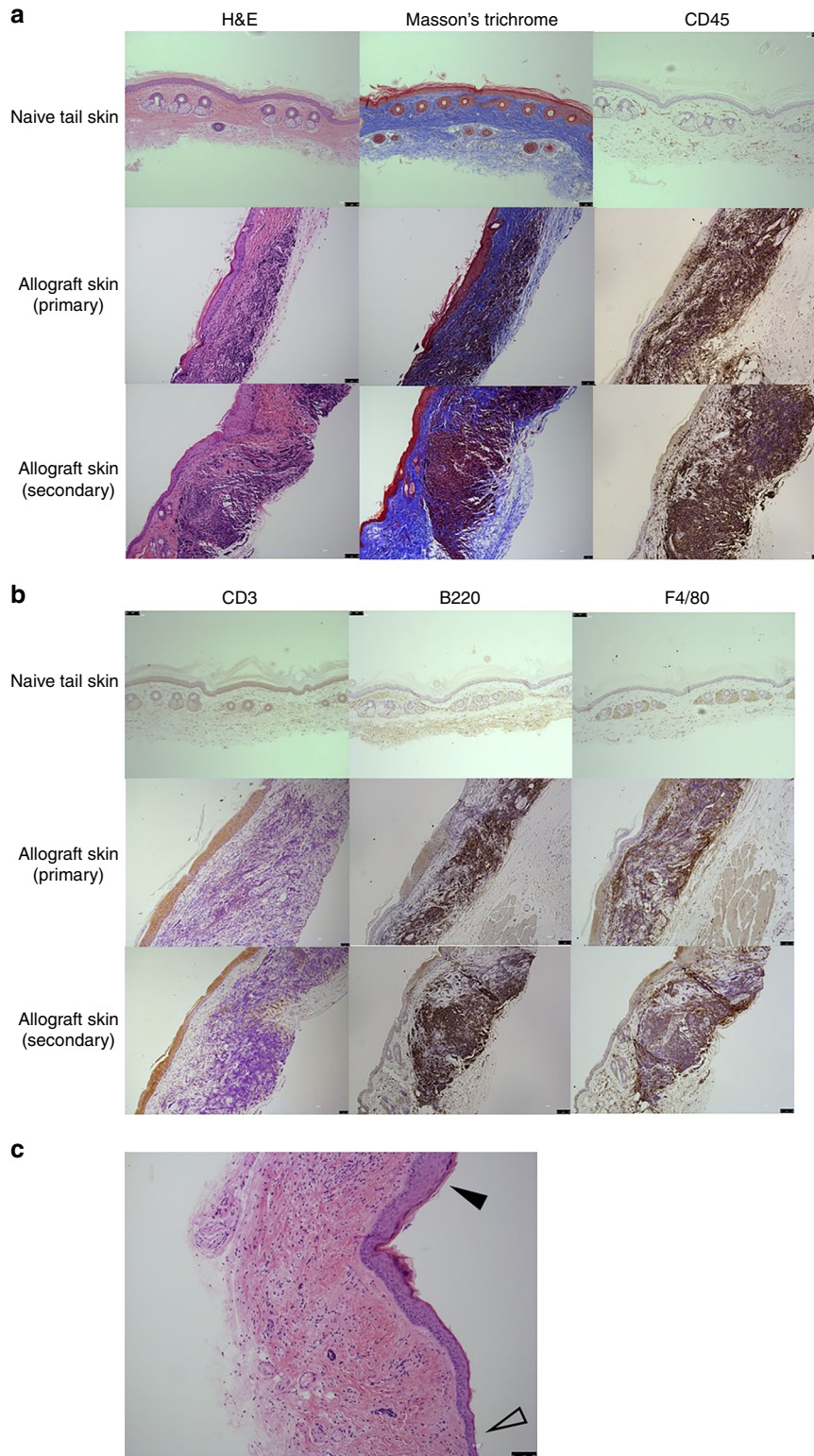

**Fig. 3** Hematopoietic cell infiltration of surviving skin allografts in CD117-ADC-conditioned recipient mice. **a**, **b** Untransplanted (naive) BALB/c tail skin (upper panels) and both primary (middle panels) and secondary (lower panels) skin allografts from the indicated mouse strains were sectioned and stained as indicated in the figure with H&E, Masson's trichrome, and CD45 (**a**) or CD3, B220, and F4/80 (**b**). Data are representative of 11 mice from two independent experiments. **c** Immune cell infiltration of surviving skin allografts from CD117-ADC conditioned mice does not extend into adjacent recipient skin. Cell infiltration at the edge of skin allografts (filled arrow head) and the surrounding recipient skin (open arrow head) is shown. Data are from H&E sections of skin and are representative of 11 mice from two independent experiments. Scale bars = 75 μm

previously reported[1]. All antibodies were injected via retro-orbital intravenous injection 6 days before BMT.

**Mice**. Male C57BL/6, BALB/c, and CBA/Ca mice 8–12 weeks old (body weight range: 26–29 grams) were obtained from The Jackson Laboratory (Bar Harbor, Maine, USA) and were maintained in our specific pathogen-free animal facility. The animal protocol was reviewed and approved by the NIAID Animal Care and Use Committee (ACUC), protocol number LMI 8E. All care and handling of animals was carried out in accordance with guidelines provided in the Guide for the Care and Use of Laboratory Animals published by the U.S. Department of Health and Human Services.

**Bone marrow and skin transplantation**. Allogeneic bone marrow transplantation was modified from a protocol described elsewhere[12,13,24]. Briefly, BALB/c bone marrow cells were obtained by flushing long bones, and were suspended in complete medium containing Ammonium-Chloride-Potassium lysing buffer to remove red blood cells. Unseparated bone marrow cells were injected intravenously into C57BL/6 recipient mice via tail vein. Unless otherwise specified, the number of total BM cells injected was either $2 \times 10^7$ or $5 \times 10^7$, which are respectively below and above the threshold for establishing hematopoietic chimerism in our system without conditioning using irradiation or cytotoxic agents. No chimerism could be established without transient immunosuppression of recipient mice. The immunosuppression protocol used in all mice in all experiments involved three consecutive 1 mg injections (days 0, 2, and 4 post-BMT) of each of the following monoclonal antibodies: rat anti-mouse CD4 (CD4 cell non-depleting; YTS 177), anti-CD40 ligand (MR1), and anti-mouse CD8 (CD8 cell depleting; YTS 169), all from BioXcell (West Lebanon, NH). In addition, two doses of 12 mg per kg of rapamycin (LC Laboratories, Woburn, MA) were administered via intraperitoneal injection at days 6 and 30 post-transplantation[24].

Where indicated, skin transplantation was performed as previously described by grafting full thickness tail skin measuring $1 \times 1$ cm on the two lateral flanks, one for the BALB/c donor type and the other for the CBA/Ca third party[24]. Grafts were observed daily after the removal of the bandage at day 7 post-transplantation and were considered rejected when complete loss of viable donor epithelium had occurred.

**Flow cytometry and analysis of chimerism**. Leukocytes were isolated from the blood, thymus, spleen, lymph node, BM, liver, and lung from mice that had received BM transplantation. After lysing red blood cells, the cells were washed and incubated for 20 min with Fc blocking antibody (Clone 93, 1:50 dilution, BioLegend, San Diego, CA) followed by the relevant antibody combinations (1:50 dilution) and FACS staining buffer for 30 min at 4 °C. Mixed chimerism was assessed in multiple lineages and compartments by staining cells with the following antibodies obtained from BioLegend (San Diego, CA) or BD Bioscience (San Jose, CA): H2-K$^d$ FITC(SF1-1.1), CD3-PB (17A2), CD11b-APC (M1/70), CD19-APC-Cy7 (6D5), TER119-Alexa Fluor 700 (Ter-119), Gr-1-PE-Cy7 (RB6-8C5), CD49b-PE (DX5), SCA-1-APC-Cy7 (D7), Flt3/CD135-APC (A2F10), CD127/IL-7 alpha-Brilliant Violet 605 (A7R34), CD117/c-kit-APC (2B8), lineage cocktail-PB (17A2/RB6-8C5/RA3-6B2/Ter-119/M1/70), CD150-PE-Cy7 (TC15-12F12.2), and CD34-biotin (RAM34) plus streptavidin-PE. All data were collected using an LSRII flow cytometer (BD Biosciences) and analyzed with FlowJo$^{TM}$ 10 software (version 10.2; Treestar, Ashland, OR) using previously published subset definitions[14].

**Histological analysis of skin allografts**. Skin allografts and the adjacent recipient skin were immediately isolated from euthanized recipients at the terminal time-point and fixed overnight in 2% paraformaldehyde, the samples were then dehydrated and embedded in paraffin. Paraffin blocks were processed into 5-μm sections, then were deparaffinized and subjected to H&E, Masson's trichrome and immunohistochemical staining. For immunohistochemistry, deparaffinized samples were incubated overnight at room temperature with the following primary antibodies after heat-induced antigen retrieval (90 °C, 20 min): anti-CD45, CD3, B220, and F4/80. The slides were then counterstained with a biotinylated antibody and evaluated with a peroxidase-DAB coloring system.

**Statistics**. Data were analyzed using unpaired parametric $t$-tests with Welch's correction (two-tailed) using Prism 7.02 (GraphPad Software) and are presented as the mean ± SEM of summary data (the data were approximately normally distributed). For survival curve analysis, Logrank and Gehan-Breslow-Wilcoxon tests were performed.

**Reporting Summary**. Further information on experimental design is available in the Nature Research Reporting Summary linked to this article.

## Data availability
All data generated and analyzed during this study are included in this published article (and its supplementary information files). A Source Data file is provided that contains data underlying Fig. 1B through 1F, Fig. 2B and supplementary figures 1, 2

and 7. The data that support the findings of this study are available from the corresponding author upon reasonable request.

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

## Acknowledgements

We thank Dr. Xin Xu of the Laboratory of Molecular Immunology, NIAID, NIH for technical assistance. This work was supported by the Division of Intramural Research, National Institute of Allergy and Infectious Diseases (Project Number AI000615-25), NIH. A.C. was supported by a Potter Fellowship to the Boston Children's Hospital Trust. D.J.R. is supported by grants from the NIH (RO1HL107630, R00AG029760, and UO1DK072473-01) as well as grants from The Leona M. and Harry B. Helmsley Charitable Trust, The New York Stem Cell Foundation, The Harvard Stem Cell Institute, and the American Federation for Aging Research.

## Author contributions

Z.L. designed the research, performed the experiments, interpreted the data and wrote the article. A.S. performed the experiments and interpreted the data. A.C., D.J.R., and P.M.M. designed the research, interpreted the data, and wrote the article.

## Additional information

**Competing interests:** Z.L., A.C., D.J.R., and P.M.M. are listed as inventors on a patent application disclosing CD117 antibody-drug-conjugates as a conditioning agent in allotransplantation filed with the US Patent and Trademark Office. Additional disclosures for A.C.: inventor, US patent applications (US 12/447,634; US 14/536,319; US 15/025,222; and US 15/148,837); Third Rock Ventures: Consultancy; GV: Salary; Global Blood Therapeutics: Equity Ownership, Consultancy; Editas Medicines: Equity Ownership, Patents & Royalties; Magenta Therapeutics: Equity Ownership, Patents & Royalties; Forty Seven Inc: Patents & Royalties, Beam Therapeutics: Equity Ownership, Consultancy. Additional disclosures for D.J.R.: Inventor, US patent application (US 14/509,787; US 15/148,837); Moderna Therapeutics: Equity Ownership, Patents & Royalties; Intellia Therapeutics: Equity Ownership, Patents & Royalties, Consultant; Vor Biopharma, Equity Ownership, Consultant; Magenta Therapeutics: Equity Ownership, Patents & Royalties; Stelexis Therapeutics: Equity Ownership, Consultant, Director; Convelo Therapeutics: Equity Ownership, Consultant, Director. The remaining authors declare no competing interests.

