## [Peer Review File · Nature Communications]

Reviewers' comments:

Reviewer #1 (Remarks to the Author):

The authors have not performed any additional experiments but just attempted to argue against my comments. Instead of performing a third repeat of experiments they have basically stated that 2 times performing an experiment is sufficient.

In my view the manuscript has not improved and does not meet the quality criteria for publication in Nature Communications.

Reviewer #2 (Remarks to the Author):

General comments:

This manuscript by Li, Czechowicz and colleagues describes a new approach allowing robust hematopoietic chimerism and (donor-specific) allograft tolerance after CD117-ADC-mediated conditioning in mismatched allotransplantation. This is an interesting proof-of-concept demonstration, which is clearly relevant for possible future translational applicability.

Major points:

- abstract: "...donor specific antigenic tolerance the holy grail of allotransplantation": This reviewer agrees with the other reviewer, this should be reworded.
- Indeed the level of blood chimerism seems to decrease over time. How do the authors explain this?
- This reviewer thinks it is important that the authors give a bit more perspective what needs to be done to implement this as a promising clinical strategy and in which setting they envision to use it first.
- "CD45.2-ADC conditioning failed to induce donor chimerism and resulted in death in 4 out of 11 mice within the first week after BMT."  Technically, this conditioning regimen (see other paper) should also have wiped out the endogenous HSC compartment, so it is difficult to understand why this did not work here. Probably, this needs more explanation than just the toxicity argument.
- In addition to HSC, CD117 (c-kit) is also expressed on other cell types, incl. hematopoietic progenitors, mast cells, germ (stem) cells, melanocytes and intestinal pacemaker cells. This is relevant information.
- "heavy mononuclear cell infiltration of unclear significance in the deep dermis of the surviving primary and secondary donor-type BALB/c tail skin allografts"  Is this compatible with a diagnosis of a beginning GvHD? Please comment.

Minor points:

- ref. 12: This only refers to the EMBL European Bioinformatics Institute (EBI). More details should be provided here.
- The authors observed invading B cells and macrophages in the skin histologies. Maybe it would be worth to follow up on this observation (in future studies) as for both cell types regulatory phenotypes have been described.

Reviewer #1

Point 1: The authors have not performed any additional experiments but just attempted to argue against my comments. Instead of performing a third repeat of experiments they have basically stated that 2 times performing an experiment is sufficient. In my view the manuscript has not improved and does not meet the quality criteria for publication in Nature Communications.

Response: *We performed the experiments in the same allogeneic strain combination twice and observed strong, well-controlled evidence of high level hematopoietic chimerism with acceptance of skin allografts in both with a clean background using five different control conditions: that is, no hematopoietic chimerism and 100% skin allotransplant rejection in the negative controls. Moreover, the animals were observed for chimerism and skin rejection for up to 500 days post BM transplantation. The cumulative total of animals under study was large given the effect size, and differences were highly statistically significant ($p < 0.0001$). Thus, we maintain that the results are robust and do not require a third repeat experiment to substantiate the claims made. Moreover, a third experiment would delay publication of this paper by 1.5 years. We feel that the results are sufficiently important from the point of view of clinical translation that the scientific community should know about them now, in order to begin together the hard work of clinical translation and further understanding of this system.*

Reviewer #2

General comments:

This is an interesting proof-of-concept demonstration, which is clearly relevant for possible future translational applicability.

Response: *We agree with the reviewer's comment and are eager to share this important work with the scientific community.*

Major points:

Point 1: abstract: "...donor specific antigenic tolerance the holy grail of allotransplantation": This reviewer agrees with the other reviewer, this should be reworded.

Response: *We have reworded the statement in the Abstract.*

Point 2: Indeed the level of blood chimerism seems to decrease over time. How do the authors explain this?

Response: *There does appear to be some waning of total donor chimerism in the peripheral blood, but only in a minority of animals and only at very late time points (3 of 8 in Expt 1 shown in Figure 1 and 0 of 4 in expt 2 in Supplemental Figure 1, not counting the 1 animal in each experiment that did not become chimeric). We have stated this in the Results section. It is important to note that the transplanted mice were followed for almost two years post-transplantation, near their life expectancy, and that all but one of those who developed chimerism in the blood were still chimeric at the time of sacrifice. Additionally, even those with the lowest levels of mixed chimerism in the blood on the day of sacrifice still had high*

levels of chimerism in secondary lymphoid organs (Figure 1D) with intact skin allografts. Additional work beyond the scope of this study will be needed to increase and optimize the level of chimerism we have demonstrated in this proof of principle CD117 ADC-conditioned recipient study, but despite this we believe these results are highly meaningful as is.

Point 3: - This reviewer thinks it is important that the authors give a bit more perspective what needs to be done to implement this as a promising clinical strategy and in which setting they envision to use it first.

Response: *Additional work will be needed to optimize the approach by systematically studying and potentially modifying the immunosuppression regimen, the dosing schedule, the number of donor cells transplanted, the age at transplantation, different CD117 antibody versions, etc. Then, the approach will require validation in large animals such as non-human primates before clinical translation can be safely attempted. A reasonable target disease for testing the approach would be in severe sickle cell anemia patients who require only low levels of red cell chimerism for clinical benefit or patients with chronic granulomatous disease who require only low levels of neutrophil chimerism for clinical benefit. We have added these ideas to the Discussion section.*

Point 4: - "CD45.2-ADC conditioning failed to induce donor chimerism and resulted in death in 4 out of 11 mice within the first week after BMT."  Technically, this conditioning regimen (see other paper) should also have wiped out the endogenous HSC compartment, so it is difficult to understand why this did not work here. Probably, this needs more explanation than just the toxicity argument. This point was emphasized by the **Editor**, who instructed us as follows: 'In particular, please include additional empirical evidence to clarify...the ineffective CD45.3-ADC treatment, as raised by referee #2.'

Response: *We have decided to remove the negative CD45-ADC data from the paper, since they are not the focus of the paper and are not needed to control for the positive effects of CD117-ADC. The key negative controls are the CD117 antibody unconjugated to saporin and the isotype control antibody-ADC, not CD45-ADC. Moreover, there is no prior expectation that CD45-ADC should promote chimerism and tolerance in an allogeneic setting, since there is no precedent in the literature on this point. Explaining our negative results with CD45-ADC would be either trivial and unimportant (i.e. the agent was inactive, didn't deplete HSCs) or an entire project beyond the scope of this report (the agent is active, but doesn't promote chimerism/tolerance: why?). The key point, and the focus of the paper, is that CD117-ADC is a novel immunotoxin that does strongly promote hematopoietic chimerism and skin allotransplantation, and that this is validated with appropriate negative controls.*

Point 5: - In addition to HSC, CD117 (c-kit) is also expressed on other cell types, incl. hematopoietic progenitors, mast cells, germ (stem) cells, melanocytes and intestinal pacemaker cells. This is relevant information.

Response: *We have added this important point to the Results section.*

Point 6: - "heavy mononuclear cell infiltration of unclear significance in the deep dermis of the surviving primary and secondary donor-type BALB/c tail skin allografts"  Is this compatible with a diagnosis of a beginning GvHD? Please comment.

Response: *The impact of this infiltrate is unclear. This could be incipient GvHD, or not, as we point out now in the Results section. We only know at present that the skin is anatomically intact for as long as we have followed the chimeric skin-transplanted animals (up to 200 days post-skin transplant). Importantly, the infiltrate is limited to the graft, and does not involve the adjacent host skin as would be expected if the process was GvHD. Future long-term experiments beyond the scope of this study will be needed to determine whether the cellular infiltrate is regulatory in nature or inflammatory.*

Minor points:

Point 1: ref. 12: This only refers to the EMBL European Bioinformatics Institute (EBI). More details should be provided here.

Response: *We now provide a complete reference.*

Point 2: - The authors observed invading B cells and macrophages in the skin histologies. Maybe it would be worth to follow up on this observation (in future studies) as for both cell types regulatory phenotypes have been described.

Response: *We agree and have planned to investigate this point in future studies.*

Please note that although not requested, we have also added a Supplemental Figure 3 showing evidence of transient elevations of liver enzymes, as was also described in the co-submitted manuscript. This was the only toxicity observed with this regimen. We also added a Supplemental file showing our FACS gating strategy, as required by the Nature Communications submissions process.

Thank you again for your consideration and interest in our work, which we hope with these revisions will be now acceptable for publication in Nature Communications.

Reviewer #1

Point 1: The authors have not performed any additional experiments but just attempted to argue against my comments. Instead of performing a third repeat of experiments they have basically stated that 2 times performing an experiment is sufficient. In my view the manuscript has not improved and does not meet the quality criteria for publication in Nature Communications.

Response: *We performed the experiments in the same allogeneic strain combination twice and observed strong, well-controlled evidence of high level hematopoietic chimerism with acceptance of skin allografts in both with a clean background using five different control conditions: that is, no hematopoietic chimerism and 100% skin allotransplant rejection in the negative controls. Moreover, the animals were observed for chimerism and skin rejection for up to 500 days post BM transplantation. The cumulative total of animals under study was large given the effect size, and differences were highly statistically significant ($p < 0.0001$). Thus, we maintain that the results are robust and do not require a third repeat experiment to substantiate the claims made. Moreover, a third experiment would delay publication of this paper by 1.5 years. We feel that the results are sufficiently important from the point of view of clinical translation that the scientific community should know about them now, in order to begin together the hard work of clinical translation and further understanding of this system.*

Reviewer #2

General comments:

This is an interesting proof-of-concept demonstration, which is clearly relevant for possible future translational applicability.

Response: *We agree with the reviewer's comment and are eager to share this important work with the scientific community.*

Major points:

Point 1: abstract: "...donor specific antigenic tolerance the holy grail of allotransplantation": This reviewer agrees with the other reviewer, this should be reworded.

Response: *We have reworded the statement in the Abstract.*

Point 2: Indeed the level of blood chimerism seems to decrease over time. How do the authors explain this?

Response: *There does appear to be some waning of total donor chimerism in the peripheral blood, but only in a minority of animals and only at very late time points (3 of 8 in Expt 1 shown in Figure 1 and 0 of 4 in expt 2 in Supplemental Figure 1, not counting the 1 animal in each experiment that did not become chimeric). We have stated this in the Results section. It is important to note that the transplanted mice were followed for almost two years post-transplantation, near their life expectancy, and that all but one of those who developed chimerism in the blood were still chimeric at the time of sacrifice. Additionally, even those with the lowest levels of mixed chimerism in the blood on the day of sacrifice still had high*

levels of chimerism in secondary lymphoid organs (Figure 1D) with intact skin allografts. Additional work beyond the scope of this study will be needed to increase and optimize the level of chimerism we have demonstrated in this proof of principle CD117 ADC-conditioned recipient study, but despite this we believe these results are highly meaningful as is.

Point 3: - This reviewer thinks it is important that the authors give a bit more perspective what needs to be done to implement this as a promising clinical strategy and in which setting they envision to use it first.

Response: *Additional work will be needed to optimize the approach by systematically studying and potentially modifying the immunosuppression regimen, the dosing schedule, the number of donor cells transplanted, the age at transplantation, different CD117 antibody versions, etc. Then, the approach will require validation in large animals such as non-human primates before clinical translation can be safely attempted. A reasonable target disease for testing the approach would be in severe sickle cell anemia patients who require only low levels of red cell chimerism for clinical benefit or patients with chronic granulomatous disease who require only low levels of neutrophil chimerism for clinical benefit. We have added these ideas to the Discussion section.*

Point 4: - "CD45.2-ADC conditioning failed to induce donor chimerism and resulted in death in 4 out of 11 mice within the first week after BMT."  Technically, this conditioning regimen (see other paper) should also have wiped out the endogenous HSC compartment, so it is difficult to understand why this did not work here. Probably, this needs more explanation than just the toxicity argument. This point was emphasized by the **Editor**, who instructed us as follows: 'In particular, please include additional empirical evidence to clarify...the ineffective CD45.3-ADC treatment, as raised by referee #2.'

Response: *We have decided to remove the negative CD45-ADC data from the paper, since they are not the focus of the paper and are not needed to control for the positive effects of CD117-ADC. The key negative controls are the CD117 antibody unconjugated to saporin and the isotype control antibody-ADC, not CD45-ADC. Moreover, there is no prior expectation that CD45-ADC should promote chimerism and tolerance in an allogeneic setting, since there is no precedent in the literature on this point. Explaining our negative results with CD45-ADC would be either trivial and unimportant (i.e. the agent was inactive, didn't deplete HSCs) or an entire project beyond the scope of this report (the agent is active, but doesn't promote chimerism/tolerance: why?). The key point, and the focus of the paper, is that CD117-ADC is a novel immunotoxin that does strongly promote hematopoietic chimerism and skin allotransplantation, and that this is validated with appropriate negative controls.*

Point 5: In addition to HSC, CD117 (c-kit) is also expressed on other cell types, incl. hematopoietic progenitors, mast cells, germ (stem) cells, melanocytes and intestinal pacemaker cells. This is relevant information.

Response: *We have added this important point to the Results section. Specifically, we now state, "As in the syngeneic bone marrow transplantation model (Czechowicz et al, co-submitted manuscript), we observed that CD117-ADC conditioning in the allotransplantation setting did not cause any gross toxicity and mice appeared healthy with preserved coat color post treatment, although similar transient elevations of liver function tests was observed (Supplementary Figure 3). Since CD117 is not just expressed on hematopoietic stem and progenitor cells, but also on mast cells, melanocytes, rare intestinal cells and germ cells, once a clinical immune suppression regimen is optimized, further experiments will be needed to examine whether CD117-ADC conditioning in the allogeneic setting affects the function of these and other CD117 expressing cells. However, when used as a sole treatment no effects have been observed to date (Czechowicz et al, co-submitted manuscript)." Specifically, we now additionally show the lack of graying of fur post CD117-ADC treatment, suggesting that melanocyte*

toxicity is likely lower compared to competitive antagonistic anti-CD117 antibody approaches (Czechowicz et al, Fig S5a). Detailed assessment of toxicity and effects on other cell types will be critical once optimized clinical-grade CD117-ADCs are developed, but in the interim we believe further experimentation in these proof-of-concept settings is of low yield.

Point 6: - "heavy mononuclear cell infiltration of unclear significance in the deep dermis of the surviving primary and secondary donor-type BALB/c tail skin allografts"  Is this compatible with a diagnosis of a beginning GvHD? Please comment.

Response: *The impact of this infiltrate is unclear. This could be incipient GvHD, or not, as we point out now in the Results section. We only know at present that the skin is anatomically intact for as long as we have followed the chimeric skin-transplanted animals (up to 200 days post-skin transplant). Importantly, the infiltrate is limited to the graft, and does not involve the adjacent host skin as would be expected if the process was GvHD. Future long-term experiments beyond the scope of this study will be needed to determine whether the cellular infiltrate is regulatory in nature or inflammatory.*

Minor points:

Point 1: ref. 12: This only refers to the EMBL European Bioinformatics Institute (EBI). More details should be provided here.

Response: *We now provide a complete reference.*

Point 2: - The authors observed invading B cells and macrophages in the skin histologies. Maybe it would be worth to follow up on this observation (in future studies) as for both cell types regulatory phenotypes have been described.

Response: *We agree and have planned to investigate this point in future studies.*

Please note that although not requested, we have also added a Supplemental Figure 3 showing evidence of transient elevations of liver enzymes, as was also described in the co-submitted manuscript. This was the only toxicity observed with this regimen. We also added a Supplemental file showing our FACS gating strategy, as required by the Nature Communications submissions process.

Thank you again for your consideration and interest in our work, which we hope with these revisions will be now acceptable for publication in Nature Communications.